# Identification Mechanism of BACE1 on Inhibitors Probed by Using Multiple Separate Molecular Dynamics Simulations and Comparative Calculations of Binding Free Energies

**DOI:** 10.3390/molecules28124773

**Published:** 2023-06-15

**Authors:** Yiwen Wang, Fen Yang, Dongliang Yan, Yalin Zeng, Benzheng Wei, Jianzhong Chen, Weikai He

**Affiliations:** 1School of Information Science and Electrical Engineering, Shandong Jiaotong University, Jinan 250357, China; 15550053180@163.com (Y.W.); fen_yang0223@163.com (F.Y.); 15367294293@163.com (Y.Z.); 2School of Aeronautics, Shandong Jiaotong University, Jinan 250357, China; 3School of Science, Shandong Jiaotong University, Jinan 250357, China; 4Center for Medical Artificial Intelligence, Shandong University of Traditional Chinese Medicine, Qingdao 266112, China; wbz99@sina.com

**Keywords:** BACE1, Alzheimer’s disease, molecular dynamics simulations, SIE, MM-GBSA

## Abstract

β-amyloid cleaving enzyme 1 (BACE1) is regarded as an important target of drug design toward the treatment of Alzheimer’s disease (AD). In this study, three separate molecular dynamics (MD) simulations and calculations of binding free energies were carried out to comparatively determine the identification mechanism of BACE1 for three inhibitors, 60W, 954 and 60X. The analyses of MD trajectories indicated that the presence of three inhibitors influences the structural stability, flexibility and internal dynamics of BACE1. Binding free energies calculated by using solvated interaction energy (SIE) and molecular mechanics generalized Born surface area (MM-GBSA) methods reveal that the hydrophobic interactions provide decisive forces for inhibitor–BACE1 binding. The calculations of residue-based free energy decomposition suggest that the sidechains of residues L91, D93, S96, V130, Q134, W137, F169 and I179 play key roles in inhibitor–BACE1 binding, which provides a direction for future drug design toward the treatment of AD.

## 1. Introduction

Alzheimer’s disease (AD) is the most common chronic neurodegenerative disease, and is characterized by early clinical manifestations of memory impairment [1,2]. As the disease progresses, patients endure various neuro-psychiatric symptoms, including language disorders, emotional disturbances, motor impairments, visuospatial skill impairments, and even personality and behavioral changes, which seriously affect social, occupational, and daily life, leading to a gradual loss of body functions and ultimate death [3]. Previous reports indicated that the development and deterioration of AD are closely related to two major pathological features, involving amyloid plaques containing the amyloid-β (Aβ) peptide and neurofibrillary tangles (NFTs) rich in tau [4,5,6]. The formation of the Aβ peptide with a 38−43 amino acid length and subsequent aggregation is the culprit responsible for AD, as it blocks the transmission between neurons, ultimately leading to neuronal death [7,8]. Toxic amyloid-β peptides are yielded through the procession of the β-amyloid precursor protein (APP) with β-amyloid cleaving enzyme 1 (BACE1) and γ-secretase [9,10]. Therefore, effective inhibition of the activity of BACE1 is a useful approach toward the treatment of AD.

In recent years, the pathway for inhibiting the activity of BACE1 through the design of small molecules has been paid widespread attention [11,12,13,14,15,16]. Several potential BACE1 inhibitors have been designed by different groups and tested in clinical trials [17,18,19]. Unfortunately, the efficiency of inhibitors is greatly limited due to their side effects. To resolve this issue, many works, involved in obtaining insights into the factors affecting inhibitor–BACE1 binding, have been performed to determine efficient schemes for the design of potent inhibitors [17,20,21,22,23,24]. Currently, it is still a great challenge to designing potent BACE1 inhibitors toward the treatment of AD. Although some BACE1 inhibitors are proposed, the molecular mechanism of inhibition of BACE1 activity is not insufficient, which imposes a heavy limit to the development of clinically available BACE1 inhibitors. Therefore, it is highly necessary to explore the mechanisms of binding between inhibitors and BACE1 at atomic levels for designing efficient BACE1 inhibitors toward the treatment of AD.

Multiple computational technologies, such as molecular dynamics (MD) simulations [25,26,27,28,29,30], calculations of binding free energies [31,32,33,34], principal component analysis, (PCA) [35,36,37,38] etc., play significant roles in investigating the atomic-level mechanism of binding between inhibitors and targets. Conventional MD simulations (cMD) are usually used to obtain conformational samplings of inhibitor–target complexes, but multiple separate MD (MSMD) simulations recently adopted by different work groups can reasonably improve the sampling efficiency of conformations [39,40,41,42,43,44,45]. Binding free energy calculations are applied to evaluate the binding ability and modes of inhibitors to targets, which are involved in molecular mechanics Poisson–Boltzmann/generalized Born surface area (MM-PB/GBSA) [46,47,48], solvated interaction energy (SIE) [49], thermodynamic integration (TI) [50,51,52] and free energy perturbation (FEP) [53,54,55,56]. Although FEP and TI methods can provide more accurate results, they are extremely time-consuming and require sufficient statistical samplings. Compared to the FEP and TI methods, MM-PB/GBSA and SIE methods can obtain fast and rational results in predictions of binding free energies. Interestingly, MD simulations and binding free energy calculations have been utilized to investigate the inhibitor–BACE1 binding mechanism [16,57,58,59,60]. For instance, Chen et al. applied Gaussian accelerated molecular dynamics simulations and MM-GBSA calculations to study the effect of pH-dependent protonation on inhibitor-BACE1 binding and their results revealed that pH-dependent protonation strongly affected the structural flexibility and correlated motions of BACE1 [61]. Hatmal and coworkers combined MD simulations and ligand–receptor contact analysis to develop valid pharmacophore models and their work rationally guided pharmacophore design [62]. Despite these successful studies, it is still of high significance to deeply investigate the molecular mechanism of inhibitor–BACE1 binding for the design of clinically available drugs for treatment of the AD.

In this work, three inhibitors, 60W, 954 and 60X [63], indicated by using their identity document (ID) in the protein data bank (PDB), were selected to explore the binding mechanism of inhibitors to BACE1 at atomic levels. The topological structure of the inhibitor–BACE1 complex and binding pocket are depicted in Figure 1A,B, respectively. The structures of 60W, 954 and 60X are separately displayed in Figure 1C–E. As shown in Figure 1C–E, three inhibitors share a similar molecular scaffold and have a small structural difference. The inhibition constants, Ki, of 60W, 954 and 60X on BACE1 are 1, 45 and 48 nM, respectively. Insights into the effect of a tiny structural difference in three inhibitors on conformational changes of BACE1 will be of importance for the design of potent inhibitors, which is the reason why we selected these three inhibitors. To achieve our goal, MSMD simulations, MM-GBSA, SIE, PCA, dynamics cross-correlation maps (DCCMs) and free energy landscapes (FELs) were combined together to clarify the identification mechanism of BACE1 for inhibitors, which includes the following contents. (1) The changes in conformations and free energy profiles were revealed through PCA and constructions of FELs, (2) binding free energies were estimated by using the MM-GBSA and SIE methods to evaluate the inhibitor–BACE1 binding ability and (3) hot interaction hotspots of inhibitors with BACE1 were identified through calculations of residue-based free energy decomposition. This work is also expected to provide useful information and theoretical guidance for the design of efficient inhibitors against BACE1.

## 2. Results and Discussion

### 2.1. Dynamics Equilibrium and Structural Fluctuation

To check the structural stability of BACE1 during three separate MD simulations, root-mean-square deviations (RMSDs) of backbone atoms in BACE1 relative to the initially optimized structures were calculated based on three separate MD trajectories and the results are provided in the Appendix A. It is noted that all four systems reached the equilibrium after 300 ns of three separate MD simulations and showed stable structural fluctuations. To better reveal the target sites, the RMSDs of the binding pocket, namely residues away from the 7 Å of the mass center of inhibitors, were calculated relative to the initially optimized structures and their probability distributions are depicted in Figure 2A. It was found that binding of inhibitors decreases the RMSDs of the BACE1 binding pocket, implying that residues around 7 Å of the mass centers of inhibitors may possibly be used as target sites of drug design toward the treatment of AD.

To examine an inhibitor’s binding-mediated impacts on the structural flexibility of BACE1, root-mean fluctuations (RMSFs) were computed by using the coordinates of the Cα atoms kept at the SMT (Figure 2C). It was found that the *apo* and bound states of BACE1 share similar flexible and rigid regions. Meanwhile, structural regions D1–D6 showed stronger flexibility and these flexible regions are possibly involved in significant functions of BACE1. To clarify the influences of inhibitors on the flexible regions of BACE1, the RMSF difference between the *apo* and bound states was also calculated by utilizing the equation $\Delta \mathrm{RMSF}=RMSF_{bound}-RMSF_{apo}$, in which ΔRMSF, $RMSF_{bound}$ and $RMSF_{apo}$ represent the RMSF difference, and the RMSFs of the bound and *apo* states (Figure 2D). The presence of the three inhibitors weakens the structural flexibility of regions D1 (residues 110–123) and D4 (residues 178–202), which makes these two regions more rigid than those of the *apo* BACE1 (Appendix A). However, the binding of the three inhibitors strengthens the structural flexibility of the regions D2 (residues 146–153) and D3 (residues 164–178) relative to that of the *apo* BACE1 (Figure 2D and Appendix A). The presence of 60W and 60X totally enhances the structural flexibility of region D5 (residues 217–245) but the binding of 954 weakens that of this region relative to the *apo* BACE1 (Figure 2D). In comparison to that in the *apo* BACE1, the binding of 60W increases the structural flexibility of region D6 (residues 363–387) while the presence of 954 and 60X slightly weakens that of this region (Figure 2D and Appendix A).

To access alterations of secondary structures for BACE1, a combination of the program CPPTRAJ and DSSP second structure analysis [64] were used to investigate the changes in the second structure of BACE1 from the *apo* and bound states in three separate MD simulations. The time evolutions of the secondary structures for the *apo*, 60W-, 954- and 60X-bound BACE1 are displayed in Figure 3A–C and Appendix A, individually. It is observed that the secondary structure of the *apo* BACE1 hardly changes through three separate MD simulations (Figure 3A,B). In comparison to the *apo* BACE1, the secondary structures of the 60W-, 954- and 60X-bound BACE1 did not generate obvious changes (Appendix A), indicating that binding of inhibitors hardly affects the stability of the secondary structures of BACE1. To understand the influences of inhibitor binding on the structure-compact components of BACE1, the gyrations of BACE1 in the *apo* and bound states were estimated based on the SMT and their probability distributions are depicted in Figure 3D. The gyration of the *apo* BACE1 was distributed at two peaks of 20.87 and 21.02 while the ones of the 60W-, 954- and 60X-associated BACE1 were separately populated at the single peaks of 21.17, 20.99 and 20.99 Å. Furthermore, the distribution shapes of gyrations for BACE1 in the three bound states totally moved toward the right with regard to the *apo* BACE1 (Figure 3D). In comparison to the *apo* BACE1, inhibitor binding only brought on a slight effect on the compact components of BACE1.

Based on the current analyses, the binding of the three inhibitors affects structural fluctuations in BACE1 but hardly changes the compact components and stability of the secondary structures for BACE1. The RMSDs of the binding pocket are obviously reduced because of inhibitor binding, implying that residues around 7 Å away from the mass center of inhibitors can be used as target sites. The difference in structures of 60W, 954 and 60X also leads to their different stability in the binding pocket of BACE1. Meanwhile, the binding of inhibitors generates an obvious effect on the structural flexibility of BACE1 and changes the flexibility of some structural regions of BACE1, which also indicates possible hot spots of inhibitor–BACE1 interactions and can provide guidance for future drug design. These current results basically agree with those of a previous work [16].

### 2.2. Conformational Changes in BACE1 and Free Energy Profiles

To clarify inhibitor-mediated changes in the correlated motions of BACE1, DCCMs were computed by means of the coordinates of the Cα atoms in BACE1 and the results are exhibited in Figure 4. The color bar coded by different colors was employed to embody the contents of the correlated motions between residues of BACE1. For the *apo* BACE1, several strongly correlated motions were observed: (1) region R1 describes strong PCMs of the N-terminal from BACE1 relative to itself (Figure 4A), (2) region R2 reflects the strong ACMs of residues 206–265 relative to residues 115–158 and region R3 characterizes strong ACMs between residues 278–331 and 146–200, (3) region R4 embodies the strong ACMs of residues 347–418 relative to the N-terminal of BACE1 and (4) region R5 describes the strong PCMs of residues 266–328 relative to themselves (Figure 4A). Compared to that in the *apo* BACE1, the binding of the three inhibitors highly weakened the PCMs occurring at regions R1 and R5 (Figure 4B,D). Meanwhile, the binding of 60W, 954 and 60X also reduced the ACMs of regions R2 and R3 with regard to those of the *apo* BACE1 (Figure 4B,D). Differently, the binding of 60W and 954 obviously abated the ACMs of region R4 in comparison to that in the *apo* BACE1 (Figure 4B,C) but the presence of 60X slightly strengthened the ACMs of region R4 (Figure 4D). The changes in correlated motions in the aforementioned regions, R1–R5, only reflect the different motion behaviors between the local structure regions and these regions are possibly involved in hot spots of the interaction of inhibitors with BACE1.

To reveal the impacts of inhibitor binding on the concerted movements of structural domains in BACE1, PCA was carried out using the CPPTRAJ in Amber 20. The first eigenvector from the PCA was visualized by means of the software VMD [65] and the results were depicted Figure 5. It can be observed that inhibitor binding has evident influences on the collective motions of α helix α1 and loop L2 (Figure 5). In the *apo* BACE1, the α1 and L2 generated a parallel concerted motion with the same direction (Figure 5A). Compared to the *apo* BACE1, the binding with 60W enhanced the concerted motion of α1 and L2 and obviously altered the direction of the concerted motion for L2 (Figure 5B). In comparison to the *apo* BACE1, the presence of 954 not only led to a completely opposite-motion direction of α1 and L2 but also inhibited the motion amplitude of L2, which created a tendency for L2 to move away from α1 (Figure 5C). With regard to the *apo* BACE1, binding with 60X not only changed the concerted-motion direction of α1 and L2 but also apparently weakened the concerted motion of L2 (Figure 5D). The binding of 60W and 954 slightly inhibited the concerted movement of loop L1 relative to that in the *apo* BACE1 but the presence of 60X slightly strengthened the concerted motion of this loop (Figure 5B,D). In addition, the binding of the three inhibitors weakened the concerted motion of loop L3 compared to that in the *apo* BACE1 (Figure 5B,D).

To unveil the free energy profiles of the BACE1 conformation changes caused by inhibitor binding, FELs were created using the projections (PC1 and PC2) of the SMT on the first two eigenvectors as reaction coordinates (RCs), and the presentative structures relating to the free energy profiles are depicted in Figure 6 and Appendix A. The projections of MD trajectories can rationally reflect conformational changes in BACE1, which is the cause for our selection of them as RCs.

In the case of *apo* BACE1, three separate MD simulations captured three free energy valleys (EVs), including EV1, EV2 and EV3 (Appendix A). According to the color bar, three EVs were located at the valley bottoms of the same depth (Appendix A). Three presentative structures of *apo* BACE1 in EV1–EV3 were superimposed together (Appendix A). The results suggest that domains D4–D6 underwent deviations from each other. Structure domains D1 and D3 generated slight deviations and D2 produced slight shifts (Figure 6B)

Compared to the *apo* BACE1, the binding of 60W and 954 only resulted in two EVs (Figure 6A,D), which is less than the number of energy states in the *apo* BACE1. This result implies that the binding of 60W and 954 induces a conformational arrangement of BACE1 relative to the *apo* BACE1. The representative structures of the 60W- and 954-bound BACE1 situated at EV1 and EV2 were superimposed together to investigate structural difference (Figure 6B,E). In comparison to the *apo* BACE1 state, the binding of 60W and 954 reduced the structural deviations of structural domains D1, D3, D4 and D5 (Figure 6B,E and Appendix A). Although the binding of 60W weakened the structural deviation of D2 and D6 relative to the *apo* BACE1 state, the association of 954 led to a great deviation of D2 and D6 (Figure 6B,E and Appendix A). As shown via the structural alignment of inhibitors 60W and 954 falling within EV1 and EV2 (Figure 6C,F), 60W and 954 produce slight shifts between the two energy states, which possibly affects the binding of these two inhibitors to BACE1. As for the 60X-bound BACE1 state, three EVs were detected throughout the entirety of the MD simulations (Figure 6G). Although the binding of 60X does not alter the number of the EVs relative to the *apo* BACE1 state, the presence of 60X enhances the energy barrier between the EV3 and other two states, EV1 and EV2, with regard to the *apo* BACE1 state (Figure 6G and Appendix A), which correspondingly increases the difficulty of the transitions between EV3, EV1 and EV2. According to the structural superimposition of the 60X-bound BACE1 trapped in EV1–EV3 (Figure 6H), except for the structural domains D1, D3, D4 and D5, the binding of 60X evidently increased the deviation of D2 and D6 within the three EVs compared to the *apo* BACE1. The structural alignment of 60X falling into the EV1–EV3 indicates that 60X undergoes slight deviations within the three energetic states (Figure 6I).

Based on the aforementioned calculations of DCCMs, PCA and analyses of FELs, the binding of inhibitors changes the correlated motions between residues, affects the concerted movements of the structural domains and the alters free energy profiles of BACE1. Some of the structural domains affected by inhibitor binding are located near the binding pocket, and hence conformational changes caused by binding of inhibitors in turn alter the activity of BACE1. In fact, several previous works also detected similar results [20,61], which are in basic agreement with our current work.

### 2.3. Comparative Calculations of Binding Free Energies

To access the binding ability of 60W, 954 and 60X to BACE1, the SIE method was applied to calculate the binding affinities of the three inhibitors to BACE1 by using 500 snapshots extracted from the equilibrated section of the three separate MD simulations, namely for the SMT, in a time interval of 1.8 ns. The calculated results are listed in Table 1. It can be observed that the rank of binding affinities predicted via the SIE method is consistent with that indicated by the experimental values, which indicates that our current free energy analyses are rational and reliable.

According to Table 1, the components of binding affinities predicted via the SIE method mainly consist of intermolecular Coulomb interactions ($\Delta E_{C}$), the van der Waals ($\Delta E_{vdW}$), reaction energy ($\Delta G^{R}$) and energy changes in the molecular surface area upon binding (γ×ΔMSA). The energy contributions favoring the binding of inhibitors are those from the van der Waals interactions between binding partners (−42.50 to −54.53 kcal·mol^−1^), the intermolecular Coulomb interactions (−12.55 to −16.18 kcal·mol^−1^) and the energy contributions relating to changes in the molecular surface (−8.05 to −10.63 kcal·mol^−1^). The reaction energies fluctuate within a range from 21.27 to 24.77 kcal·mol−1 and this component provides an unfavorable force for inhibitor bindings, which was also revealed by the previous work [49,66,67]. On the basis of Table 1, the unfavorable reaction energies of three inhibitor–BACE1 complexes are partially compensated for by the favorable intermolecular Coulomb interaction. Meanwhile, the intermolecular van der Waals interactions also contribute partial compensation to this unfavorable effect. Among the three inhibitors, 60W showed the strongest binding ability to BACE1 (−8.82 kcal·mol^−1^) while 954 had the weakest binding ability to BACE1 (−7.30 kcal·mol^−1^), which suggests that a small structure difference among the three inhibitors impacts their binding ability to BACE1.

To comparatively study the binding strength of 60W, 954 and 60X to BACE1, the MM-GBSA method was adopted to predict the binding free energies of the three inhibitor–BACE1 complexes based on 500 snapshots extracted from the equilibrated section of three separate MD simulations, namely for the SMT, in a time interval of 1.8 ns. Because of the high time costs of the entropy calculation, 100 snapshots taken from the above-mentioned 500 snapshots were employed to perform the calculation of the entropy contributions to inhibitor–BACE1 binding. The MM-GBSA calculations are possibly involved in multiple generalized Born (GB) models. To understand the influences of different GB models on the predicted results, four GB models, indicated by IGB = 1, IGB = 2, IGB = 5 and IGB = 66, were chosen to estimate the binding free energies of the three inhibitors to BACE1. The empirical parameters involved in the calculations of four GB models are given in Table 2, which includes two empirical parameters, γ and β, together with the radii types. The binding free energies and their components computed by the MM-GBSA method are listed in Table 3.

Binding free energies are mainly composed of five components, including van der Waals interactions ($\Delta E_{vdW}$), electrostatic interactions ($\Delta E_{ele}$), polar solvation free energy ($\Delta G_{gb}$), non-polar solvation free energy ($\Delta G_{surf}$) and entropy contributions (−TΔS), which are shown in Table 3. From the free energy components, $\Delta E_{vdW}$, $\Delta E_{ele}$ and $\Delta G_{surf}$ are favorable for inhibitor–BACE1 binding but $\;\Delta G_{gb}$ and −TΔS impair inhibitor–BACE1 associations (Table 3). The hydrophobic interactions ($\Delta G_{hydro}$) formed by the sum of $\Delta E_{vdW}$ and $\Delta G_{surf}$ are favorable for inhibitor–BACE1 binding. The polar interactions ($\Delta G_{pol}$) formed by the sum of $\Delta E_{ele}$ and $\Delta G_{gb}$ provide a force of a different type for inhibitor–BACE1 association. In detail, the $\Delta G_{gb}$ predicted via the models IGB = 1, IGB = 2 and IGB = 66 was unfavorable for inhibitor–BACE1 binding while that predicted by model IGB = 5 contributed favorable forces to the inhibitor–BACE1 associations (Table 3). The sum of three components, $\Delta E_{vdW}$, $\Delta E_{ele}$, and $\Delta G_{gb}$ makes up the enthalpy contributions (∆H) to inhibitor–BACE1 binding. Based on Table 3, the GB models used for calculations of MM-GBSA only produced evident impacts on polar solvation free energies and the selection of the empirical parameters γ and β obviously affected the calculations of non-polar solvation free energies. Comparing the four GB models, the GB model IGB = 5 led to the weakest polar solvation free energies for all inhibitors but IGB = 66 yielded the strongest polar solvation free energies (Table 3). Correspondingly, the GB model IGB = 5 generated the strongest enthalpy contributions to inhibitor–BACE1 association but IGB = 66 produced the weakest enthalpy contributions to inhibitor–BACE1 binding. As a result, the selection of the GB models brought on a vital impact on the predictions of inhibitor–BACE1 binding free energies.

For our used GB models, the binding free energies of 60W, 954 and 60X to BACE1 estimated with the GB model IGB = 2 were mostly close to the experimental values. Differently, the binding free energies of the three inhibitors to BACE1 calculated through the GB models IGB = 1, 5 and 66 highly deviated from the experimental results. Meanwhile, the rank for the binding free energies of 60W, 954 and 60X in the four GB models was also in good agreement with that from the experimental values, verifying that our current results are reliable and rational. Based on the aforementioned analyses, the results calculated via GB model IGB = 2 were utilized to determine the binding difference between the three inhibitors to BACE1. The electrostatic interactions of 60W and 60X with BACE1 were, respectively, strengthened by 7.98 and 5.63 kcal/mol relative to those of 954 with BACE1 but unfavorable polar solvation free energies of the 60W- and 60X-BACE1 complexes were raised by 11.42 and 3.99 kcal/mol kin comparison to those of the 954-BACE1 complex. On the whole, the polar interaction of 60W with BACE1 was increased by 3.44 kcal/mol relative to that of 954 with BACE1 while the polar interaction of 60X with BACE1 was reduced by 1.64 kcal/mol. The hydrophobic interaction of 60W with BACE1 strengthened by 13.3 kcal/mol compared to that of 954 with BACE1, but the hydrophobic interaction of 60X with BACE1 hardly changed relative to that of 954 with BACE1. As a result, the enthalpy contributions to the 60W- and 60X-BACE1 binding were improved by 9.86 and 1.63 kcal/mol relative to those of 954-BACE1 binding. In addition, the unfavorable entropy contributions to the 60W- and 60X-BACE1 binding increased by 3.92 and 0.34 kcal/mol relative to 954-BACE1 binding. In summary, the binding ability of 60W and 60X to BACE1 was strengthened by 5.94 and 1.29 kcal/mol compared to that of 954 to BACE1 (Table 3). Therefore, although the structural difference between the three inhibitors is tiny, their binding ability to BACE1 produces a bigger difference according to our current calculations, which could be due to the conformational changes caused by their binding.

Via a combination of the SIE and MM-GBSA calculations, it was found that hydrophobic interactions provide a key contribution to inhibitor–BACE1 binding, which agrees well with the results from the previous report [16]. Thus, the rational optimization of inhibitor–BACE1 hydrophobic interactions is of high significance for the successful design of clinically available inhibitors for binding with BACE1. Based on this issue, more attention should be paid to the hydrophobic interactions of inhibitors with BACE1.

### 2.4. Analyses of Inhibitor–BACE1 Interaction Networks

To obtain atomic-level insights into the interaction modes of inhibitors with BACE1, the residue-based free energy decomposition method was applied to estimate the inhibitor-residue interaction spectrum of three inhibitor–BACE1 complexes (Figure 7). The contributions from the sidechains and backbones of residues to the inhibitor–BACE1 associations are provided in Table 4. The hydrogen bonding interactions (HBIs) between inhibitors and residues of BACE1 were analyzed using the program CPPTRAJ and the results are listed in Table 5. The geometric information regarding inhibitor–residue interactions is depicted in Figure 8. Meanwhile, the probability distributions of the distances related to inhibitor–residue interactions were also calculated and the results are displayed in Figure 9.

For the 60W-BACE1 complex, 60W produced interactions stronger than −1.0 kca/mol with six residues of BACE1, including L91, D93, S96, V130, Q134 and I179 (Figure 7A,D). The three residues D93, S96 and V130 were situated near hydrophobic rings R1 and R2 of 60W (Figure 8A). Hence, D93 formed the CH-O interactions with these two rings, S96 generates the CH-π and CH-O interactions with ring R1 and V130 yields the CH-π interaction with ring R1 of 60W (Figure 8A). According to Table 4, the energetic contributions of S96 and V130 to 60W-BACE1 binding mostly arose from the sidechains of these two residues. Additionally, the carbonyl of D93 generated four HBIs with ring R2 of 60W and their occupancy was higher than 46.7% (Table 5 and Figure 8B); meanwhile, the favorable 60W-D93 interaction mainly came from the electrostatic interaction of the sidechain of D93 (Table 4). On the whole, D93, S96 and V130 provided energy contributions of −2.04, −1.13 and −1.75 kca/mol to 60W-BACE1 binding, respectively (Figure 7A,D and Table 4). The distances for the mass centers of the sidechains of V130 and S96 from those of ring R1 were respectively distributed at 4.03 and 4.03 Å (Figure 9A), which verifies the interactions of these two residues with 60W. The distance between the mass center of the carbonyl of D93 and that of ring R2 in 60W was greatest at 6.09 Å, which agrees with the weak CH-O interaction of D93 with 60W (Figure 9A). Residues Q134 and I179 were next to ring R3 of 60W and these two residues formed the CH-π interactions with ring R3 of 60W (Figure 7A,D and Figure 8A). As shown in Table 4, the van der Waals interactions of the sidechains from Q134 and I179 with ring R3 of 60W contributed the greatest forces to the 60W-BACE1 association. The distance of the carbon atom from Q134 and that of the mass center of the alkyl group in I179 from mass center of ring R3 in 60W were situated at 4.03 and 3.66 Å, respectively (Figure 9A). As a result, the two residues, Q134 and I179, separately provided the interaction energies of −2.58 and −2.26 kcal/mol for the binding of 60W to BACE1 (Figure 7A,D). The interaction energy of L91 with 60W was −1.72 kcal/mol (Figure 7A,D), which structurally stemmed from the CH-π interaction between the alkyl group of L91 and ring R4 of 60W (Figure 8A). More interestingly, the energy contribution of L91 was mainly provided by the van der Waals interactions between the sidechain of L91 and ring R4 of 60W (Table 4). The distance between the mass center of the alkyl group from L91 and that of ring R4 in 60W was great at 3.84 Å (Figure 9A), which demonstrates the existence of the CH-π interaction between 60W and L91.

With respect to the 954-BACE1 compound, five residues were involved in interactions stronger than −1.0 kcal/mol with inhibitor 954 and these residues included L91, Y132, W137, F169 and I179 (Figure 7B,D). The interaction energies of Y132 and W137 with 954 were −1.88 and −1.04 kcal/mol, individually, which structurally agree with the π–π interactions of the phenyl group in Y132 with ring R2 of 954 and of the hydrophobic ring of W137 with ring R1 of 954 (Figure 8C). The distances of the mass centers for the hydrophobic rings of Y132 and W137 from rings R2 and R1 of 954 were located at 4.87 and 5.09 Å (Figure 9B), which further supported the interactions of these two residues with 954. Based on Table 4, the energy contributions of Y132 and W137 to 954-BACE1 binding are mostly provided by the van der Waals interactions of the sidechains from Y132 and W137 with 954. The hydrophobic groups L91, F169 and I179 were located near ring R4 of 954 (Figure 8C). Therefore, the alkyl group of L91, the phenyl group of F169 and the alkyl group of I179 tend to generate the CH-π, π-π and CH-π interactions with R4 of 954. The distances of the mass centers of the sidechains in L91, F169 and I179 from those of ring R4 in 954 were respectively greatest at 4.86, 6.42 and 4.24 Å (Figure 9B), which verifies the hydrophobic nature of interactions of these three residues with 954. On the whole, L91, F169 and I179 contributed the interaction energies of −1.14, −1.44 and −1.91 kcal/mol to 954-BACE1 binding (Figure 7B,D and Table 4). More importantly, the interaction energies of L91, F169 and I179 with 954 mostly originated from the van der Waals interactions of the sidechains in these three residues with 954 (Table 4). In addition, the carbonyl group of D93 formed four HBIs with ring R3 of 954 and the occupancy of these four hydrogen bonds was higher than 46.9% (Table 5 and Figure 8D). However, D93 only provided an energy contribution of −0.77 kcal/mol (Figure 7B,D), which mainly stemmed from the electrostatic interaction between the sidechain of D93 and 954 (Table 4).

With regard to the 60X-BACE1 complex, 60X yielded interactions stronger than −1.0 kcal/mol with five residues, D93, V130, Y132, W137 and I179, in BACE1 (Figure 7C,D). The hydrophobic groups of I179 and Y132 were adjacent to ring R2 of 60X (Figure 8E), hence the formation by the alkyl group of I179 of CH-π interactions with ring R2 of 60X and the generation by the phenyl group of Y132 of the π-π interaction with the R2 of 60X (Figure 8E). The distances of the mass centers for the hydrophobic groups of Y132 and I179 from those of ring R2 in 60X were, respectively, greatest at 4.52 and 5.13 Å (Figure 9C), which further supports the interactions of Y132 and I179 with 60X. Y132 and I179 separately contribute interaction energies of −2.0 and −2.31 kcal/mol to 60X-BACE1 binding (Figure 7C,D). Furthermore, they mostly came from the van der Waals interactions of the sidechains in Y132 and I179 with ring R2 of 60X (Table 4). Two residues, V130 and W137, produced interactions of −1.28 and −1.29 kcal/mol with 60X (Figure 7C,D), which is in good agreement with the CH-π interaction of the alkyl group from V130 and the π–π interaction of the hydrophobic ring of W137 with ring R3 of 60X (Figure 8E). The distances of the mass centers for the sidechains of V130 and W137 from those of ring R3 in 60X were separately distributed at 4.32 and 6.13 Å (Figure 9C), implying the existence of the interactions of V130 and W137 with 60X. More importantly, the energy contributions of V130 and W137 to the 60X-BACE1 association were mainly provided by the van der Waals interactions of the sidechains of V130 and W137 with 60X (Table 4). Additionally, the carbonyl group of residue D93 not only produced the CH-O interactions with ring R1 of 60X but also formed four HBIs with an occupancy higher than 48.7% with ring R2 of 60X (Figure 8F and Table 5). The distance between the mass center of the carbonyl group of D93 and that of ring R2 in 60X was distributed at 4.72 Å (Figure 9C), implying the existence of the CH-O interactions of D93 with 60X. The above revealed residues are also involved in interactions of the other inhibitors with BACE1 [16,20,63], which is in agreement with our current results.

Based on the aforementioned description, three inhibitors form hydrophobic interactions with L91, S96, V130, Q134, W137, F169 and I179 and the energy contributions of these residues to inhibitor binding mostly come from the interactions of their sidechains with inhibitors. Residue D93 produces four HBIs with inhibitors and these HBIs are formed between the carbonyl group (the sidechain) of D93 and inhibitors. It is concluded that the sidechains of the above-mentioned residues play key roles in the binding of inhibitors to BACE1. More importantly, the CH-π, CH-O, π-π interactions and HBIs between the sidechains of these eight residues and inhibitors are identified as the main inhibitor–BACE1 binding modes, which should be paid special attentions. Therefore, it is of high significance to rationally optimize the interactions of inhibitors with the sidechains of key residues in BACE1 for design of efficient inhibitors toward BACE1.

## 3. Materials and Methods

### 3.1. Construction of Initial Systems

The initial structures of 60W-, 954- and 60X-BACE1 complexes were obtained from the PDB. The ID 5HDU, 5HDZ and 5HE7 respectively correspond to the 60W-, 954- and 60X-BACE1 complexes [63]. The *apo* BACE1s without the associations of inhibitors were obtained by cutting 60W from the crystal structure 5HDU. The missing residues in three crystal structures were repaired by using the program Modeller [68]. All of the crystal water and non-inhibitor molecules were deleted from the initial model. The protonated states of residues from BACE1 were checked through the program H++ 3.0 [69]. Then, the following tasks were accomplished with the help of the module Leap in Amber 20 [70,71]: (1) the force field parameters of BACE1 were assigned by employing the ff19SB force field [72], (2) three disulfide bonds were established between C176 and C380, C238 and C403, and C290 and C340, respectively, (3) an octahedral TIP3P water molecule periodic box with a buffer of 10.0 Å was constructed to solve four BACE1-related systems, and (4) counter ions were added within the systems in a 0.15 M salt environment to neutralize each system, in which the force parameters of sodium ions (Na^+^) and chloride ions (Cl^−^) were derived from the work of Joung and Cheatham [73,74]. The molecular structures of the three inhibitors, 60W, 954 and 60X, were optimized at a semiempirical AM1 level, and then, the BCC charges [75,76] were given to each atom of the inhibitors using the Antechamber module in Amber [77]. The general Amber force field (GAFF2) [78,79] was adopted to generate the force field parameters of the three inhibitors, 60W, 954 and 60X.

### 3.2. MD Simulations

To remove possible high-energy contacts between atoms formed during the initial process of four simulated systems and relieve the instability of the systems, two-step energy minimizations were implemented before a real MSMD simulation, composed of a 50,000-step steepest-descent optimization and a 50,000-step conjugate-gradient one. Subsequently, all systems were provided a slow heating process from 0 to 300 k in 1 ns in the canonical ensemble (NVT), in which all non-hydrogen atoms in BACE1 and the inhibitors were constrained to a weak harmonic restriction of 2 kcalmol^−1^‧Å^2^. Then, a 2 ns equilibrium phase was executed on four BACE1-related systems at 300 K under the isothermal−isobaric ensemble (NPT) to further optimize the systems. Finally, 600 ns MD simulations were conducted on each system to deeply relax the systems. The above mentioned simulation processes were repeatedly performed three times, and the initial atomic velocities were produced by means of the Maxwell distribution. As a result, three separate MD simulations were completed. Through the aforementioned simulation stages, the Langevin thermostat [80] was utilized to adjust the system temperature and meanwhile the collision frequency was set as 2 ps^−1^. The shake algorithm [81] was applied to constrain all the chemical bonds involved in the hydrogen atoms. The long-range electrostatic interactions between atoms were estimated with the particle mesh Ewald algorithm [82] with a cutoff value of 9 Å. At the same time, this cutoff was also employed to calculate van der Waals interactions between atoms. The equilibrium phases of three separate MD trajectories were connected in a single MD trajectory (SMT) to facilitate the post-processing analysis. For this work, all simulations were performed by employing the program pmemd.cuda in Amber 20 [83,84].

### 3.3. Calculations of Solvated Interaction Energy

The SIE method can be used to quickly and rationally predict the binding free energies of inhibitors to targets. The SIE function for calculating inhibitor–BACE1 binding free energy can be expressed with the following equation, Equation (1):(1)$$\Delta G_{bind}\left(\rho ,D_{in},\alpha ,\gamma ,C\right)=\alpha \times \left[E_{c}\left(D_{in}\right)+\Delta G^{R}+E_{vdW}+\gamma \times \Delta MSA\left(\rho \right)\right]+C$$
in which $E_{c}$ and $E_{vdW}$ indicate the intermolecular Coulomb and van der Waals interaction energies between atoms in the bound state, respectively, and they were calculated using Amber molecular mechanics force field ff19SB. The component $\Delta G^{R}$ represents the alteration in the reaction field energy caused by the binding of inhibitors to BACE1 and was calculated by solving the Poisson equation using the boundary element method BRI BEM [85,86] together with a variable-radius solvent probe [87]. The term γ×ΔMSA is used to reflect the change in free energies related to the molecular surface area caused by inhibitor binding. The parameters ρ, $D_{in}$, γ and C represent the Amber van der Waals radii’s linear scaling coefficient, the solute interior dielectric constant, the molecular surface area coefficient and a constant, respectively. The parameter α indicates the global proportionality coefficient related to the loss of conformational entropy upon binding [88]. The optimized values of the aforementioned parameters are α=0.1048, ρ=1.1, $D_{in}=2.25$, γ=0.0129 kcal·mol^−1^·Å^−2^ and C=−2.89 kcal·mol^−1^ [49]. The SIE calculations were implemented by means of the program Sietraj [67].

### 3.4. MM–GBSA Calculations

MM–PB/GBSA methods have been widely used to calculate inhibitor–target binding free energies. Hou’s group performed a series of works to evaluate the performance of these two methods [89,90,91]. According to their tests, the MM-GBSA method was selected to calculate inhibitor–BACE1 binding free energies with the following equation, Equation (2):(2)$$\Delta G_{bind}=\Delta E_{ele}+\Delta E_{vdw}+\Delta G_{gb}+\Delta G_{surf}-T\Delta S$$
where $\Delta E_{ele}$ and $\Delta E_{vdw}$, respectively, represent the electrostatic and van der Waals interactions of inhibitors with BACE1, while $\Delta G_{pol}$ and $\Delta G_{nonpol}$, respectively, indicate the polar and nonpolar contributions to the solvent-free energy of the inhibitor–BACE1 complexes. The two components $\Delta E_{ele}$ and $\Delta E_{vdw}$ were obtained from the force of Amber force field ff19SB. The term $\Delta G_{surf}$ was estimated by using the empirical equation $\Delta G_{surf}=\gamma \times \Delta SASA+\beta$, in which ΔSASA denotes the solvent-accessible surface area. $\Delta G_{pol}$ was computed using the generalized Born (GB) model [92]. For this current study, we selected different GB models that were individually represented by IGB = 1, 2, 5 and 66 [92,93] to calculate the $\Delta G_{gb}$ so that we could examine the impacts of different GB models on the calculations of binding free energies. The type of GB model and the corresponding parameters, including radius types, γ and β, are provided in Table 1. The last component, T∆S, represents the contribution of the entropic change to the binding free energies and was estimated through the mmpbsa_py_nabnmode program in Amber 20 [94]. In our current work, 500 snapshots were extracted from the SMT to compute binding free energies. Since the entropy calculation was too expensive, 100 snapshots picked from the above-mentioned 500 snapshots were utilized to calculate the entropy contributions to inhibitor–BACE1 binding.

### 3.5. Principal Component Analysis

PCA was helpful for our insights into the concerted motions of structure domains in BACE1. Hence, PCA was executed to clarify how the binding of an inhibitor impacts the concerted motions of BACE1. In this work, PCA was realized through the diagonalization in a covariance matrix, C, constructed with the coordinates of the Cα atoms in BACE1 based on Equation (3):(3)$$C=\langle \left(q_{i}-\langle q_{i}\rangle \right){\left(q_{j}-\langle q_{j}\rangle \right)}^{T}\rangle$$
where, the terms $\;q_{i}$ and $q_{j}$ are the Cartesian coordinates of the *i*th and *j*th Cα atoms in BACE1, respectively, while the terms $\langle q_{i}\rangle$ and $\langle q_{j}\rangle$ are their averaged positions on conformational ensembles recorded at the SMT. In general, this average is computed by performing a superimposition of the SMT with a referenced structure to abolish overall translations and rotations by using a least-squares fitting procedure [95]. The eigenvalues and eigenvectors stemming from the PCA are usually applied to respectively embody the fluctuation amplitude along an eigenvector and the concerted motions of structural domains. We completed the PCA through the program CPPTRAJ [96] in Amber 20 in this study.

### 3.6. Dynamics Cross-Correlation Map

DCCMs are an efficient approach to exploring the internal dynamics of targets [97,98,99]. To clarify the influence of inhibitor binding on the internal dynamics of BACE1, DCCMs were calculated using the coordinates of the Cα atoms in BACE1 saved at the SMT through the following equation, Equation (4):(4)$$C_{ij}=\frac{\langle \Delta r_{i}\cdot \Delta r_{j}\rangle }{{\left(\langle \Delta r_{i}^{2}\rangle \langle \Delta r_{j}^{2}\rangle \right)}^{1/2}}$$
in which the two components $\Delta r_{i}$ and $\Delta r_{j}$ are the displacement of the Cα atoms i and j relative to their corresponding averaged positions. The angle brackets is an indicator of ensemble averages on the snapshots kept at the SMT. The element values ($C_{ij}$) of DCCMs fluctuated at a range from −1 to 1. The positive and negative $C_{ij}$ values, respectively, correspond to the positively correlated movements (PCMs) and the anti-correlated motions (ACMs) between the Cα atoms i and j of BACE1. The color-coded bars were employed to characterize the extent of correlated motions between residues of BACE1. In this study, the calculations of DCCMs were finished by using the program in Amber 20.

## 4. Conclusions

BACE1 plays an important role in the production of the toxic amyloid–β peptides that cause ADs. Insights into the inhibitor–BACE1 binding mechanism and conformational changes of BACE1 due to inhibitor binding are significant for the development of efficient drugs targeting BACE1. Three separate MD simulations with a total simulation time of 1.8 μs, each running for 600 ns, were respectively conducted on the *apo*, 60W-, 954- and 60X-bound BACE1. The analyses of RMSDs and RMSFs verified that inhibitor binding not only affects structural stability but also changes the structural flexibility of BACE1. The results from the calculations of DCCMs and PCA indicate that inhibitor binding alters correlated motions between the structural domains and molecular dynamics behavior of BACE1. Binding free energies calculated through the SIE and MM-GBSA methods comparatively revealed that hydrophobic interactions drive inhibitor–BACE1 binding, which should be paid special attention in future drug design targeting BACE1. Among the three inhibitors used herein, 60W showed the strongest ability to bind to BACE1 and its molecular structure should provide guidance for the design of efficient inhibitors targeting BACE1. The inhibitor–residue spectrum from calculations of residue-based free energy decomposition shows that residues L91, D93, S96, V130, Q134, W137, F169 and I179 were identified as hot spots of interaction between inhibitors and BACE1. The results also suggest that the sidechains of those eight residues provide the main energy contributions for binding of inhibitors to BACE1. Thus, the optimization of the interactions of inhibitors with the sidechains from those eight residues should be paid special attention to in the drug design for the treatment of AD. This study is also expected to contribute useful information to the development of potent inhibitors for interaction with BACE1.

## Figures and Tables

**Figure 1 molecules-28-04773-f001:**
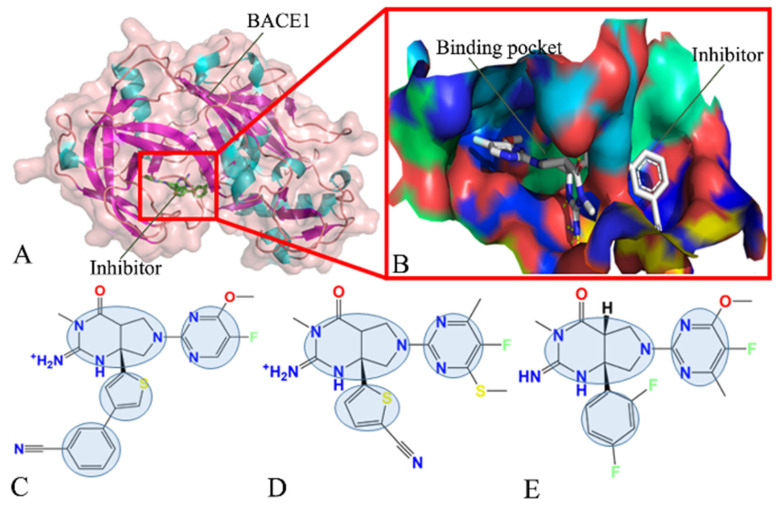
Molecular structures: (**A**) inhibitor–BACE1 complex, in which BACE1 is shown in cartoon and surface forms, while the inhibitor is displayed in stick form; (**B**) binding pocket of inhibitor to BACE1; (**C**–**E**) correspond to 60W, 954 and 60X, respectively, in which three inhibitors are displayed in line form. In this figure, the shaded areas indicate the groups of inhibitors that possibly produce hydrophobic interactions with BACE1.

**Figure 2 molecules-28-04773-f002:**
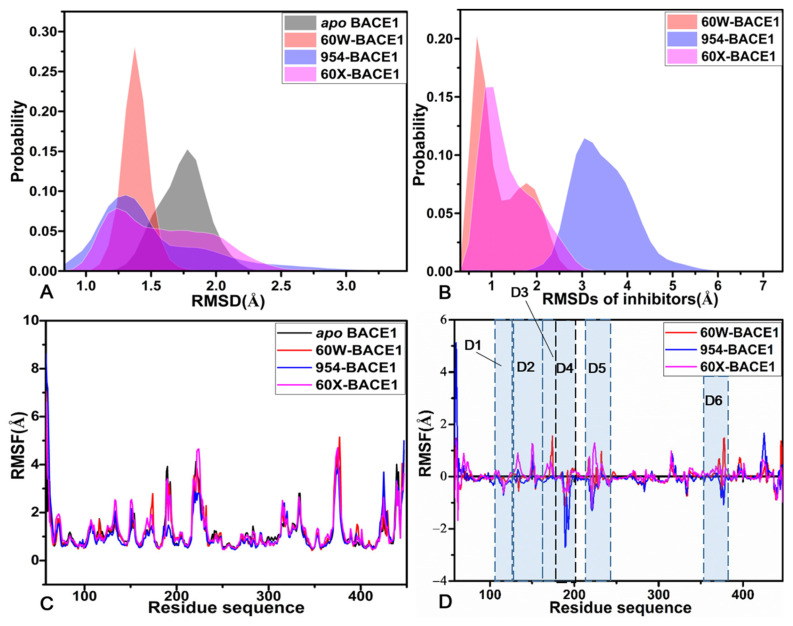
Frequency of RMSDs and RMSFs: (**A**) probability distributions of RMSDs for the binding pockets of the *apo* BACE1, 60W-, 954- and 60X-bound ones, (**B**) probability of RMSDs for three inhibitors, (**C**) RMSFs of the *apo* and bound states of BACE1 and (**D**) the difference in RMSFs between the *apo* BACE1 and the inhibitor-bound ones.

**Figure 3 molecules-28-04773-f003:**
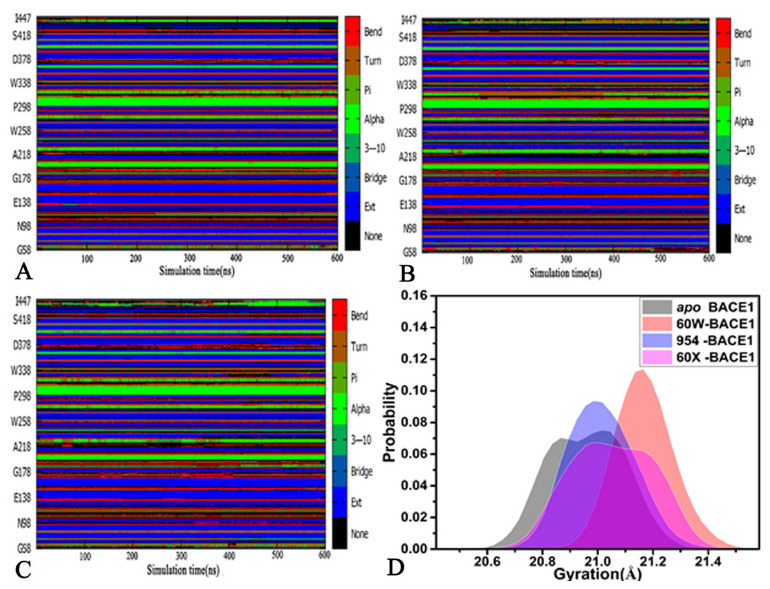
Stability of secondary structures and structure-compact components of BACE1; (**A**) time evolution of secondary structure for the *apo* BACE1 in simulation 1, (**B**) time evolution of secondary structure for the *apo* BACE1 in simulation 2, (**C**) time evolution of secondary structure for the *apo* BACE1 in simulation 3 and (**D**) the frequency distribution of the BACE1 gyration.

**Figure 4 molecules-28-04773-f004:**
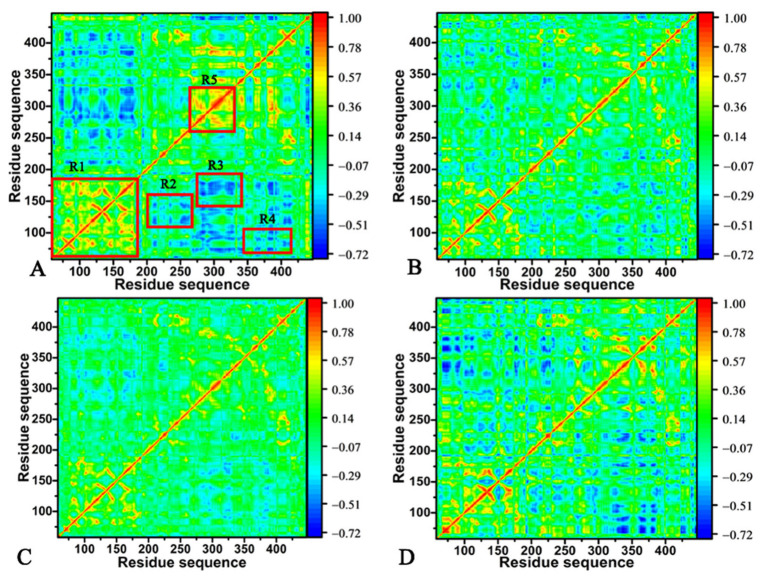
DCCMs calculated using the coordinates of the Cα atoms from BACE1; (**A**) the *apo* BACE1, (**B**) the 60W-bound BACE1, (**C**) the 954-bound BACE1 and (**D**) the 60X-bound BACE1. In this figure, the color bar is used to reflect the extents of correlated motions between the regions of BACE1.

**Figure 5 molecules-28-04773-f005:**
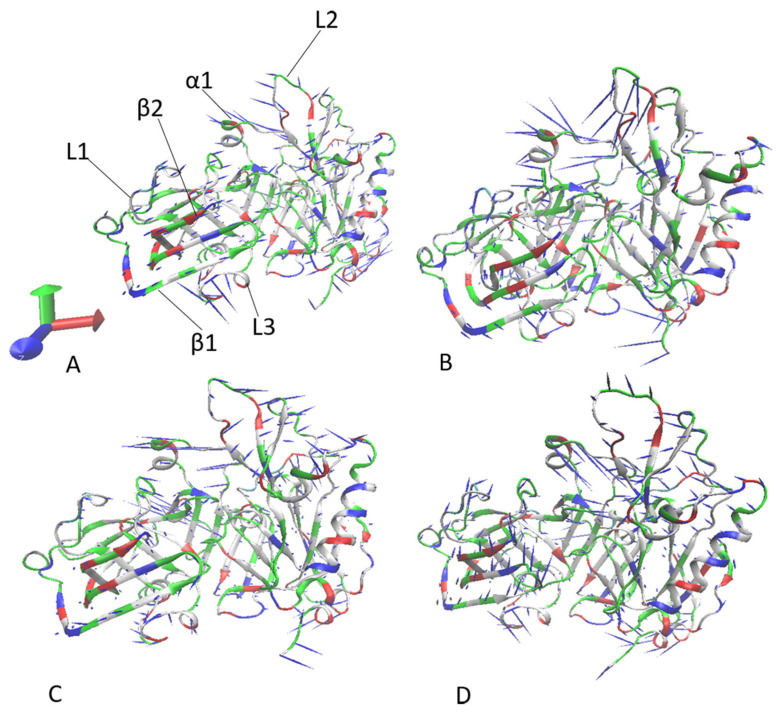
Concerted motions of structural domains from BACE1 revealed by PCA: (**A**) the *apo* BACE1, (**B**) the 60W-bound BACE1, (**C**) the 954-bound BACE1 and (**D**) the 60X-bound BACE1.

**Figure 6 molecules-28-04773-f006:**
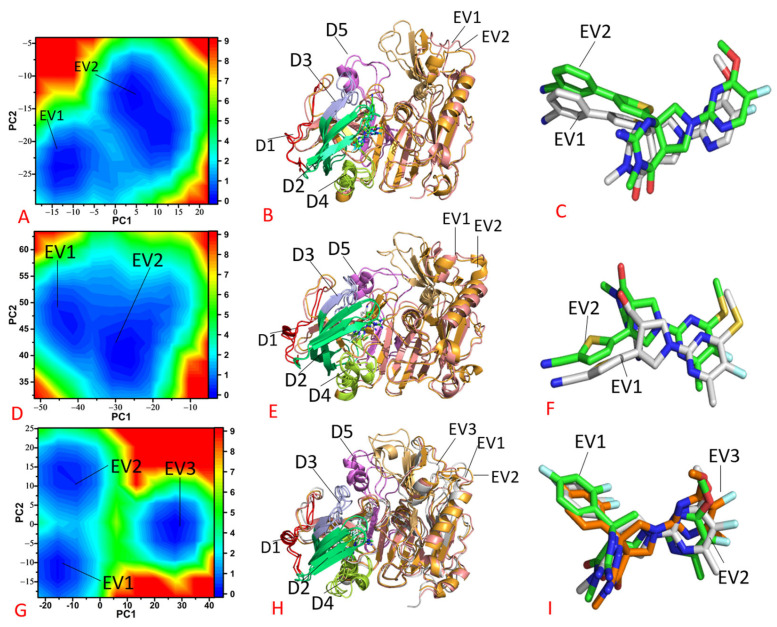
Free energy landscapes and the representative structures; (**A**) free energy landscape of the 60W-bound BACE1, (**B**) structural superimpositions of the 60W-bound BACE1 located at EV1 and EV2, (**C**) structural alignment of 60W falling into EV1 and EV2, (**D**) free energy landscape of the 954-bound BACE1, (**E**) structural alignment of the 954-bound BACE1 situated at EV1 and EV2, (**F**) structural superimposition of 954 located at EV1and EV2, (**G**) free energy landscape of the 60X-bound BACE1, (**H**) superimposition of the 60X-bound BACE1 and (**I**) structural alignment of 60X trapped at EV1–EV3.

**Figure 7 molecules-28-04773-f007:**
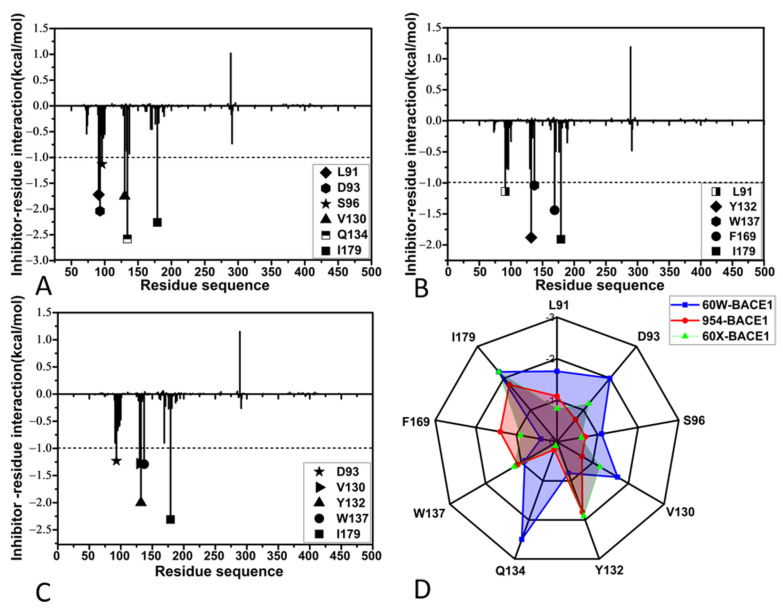
Interactions of inhibitors with BACE1; (**A**) interaction spectrum of 60W with separate residues of BACE1, (**B**) interaction spectrum of 954 with each residue of BACE1, (**C**) interaction spectrum of 60X with separate residues of BACE1 and (**D**) key residues in inhibitor–BACE1 interactions.

**Figure 8 molecules-28-04773-f008:**
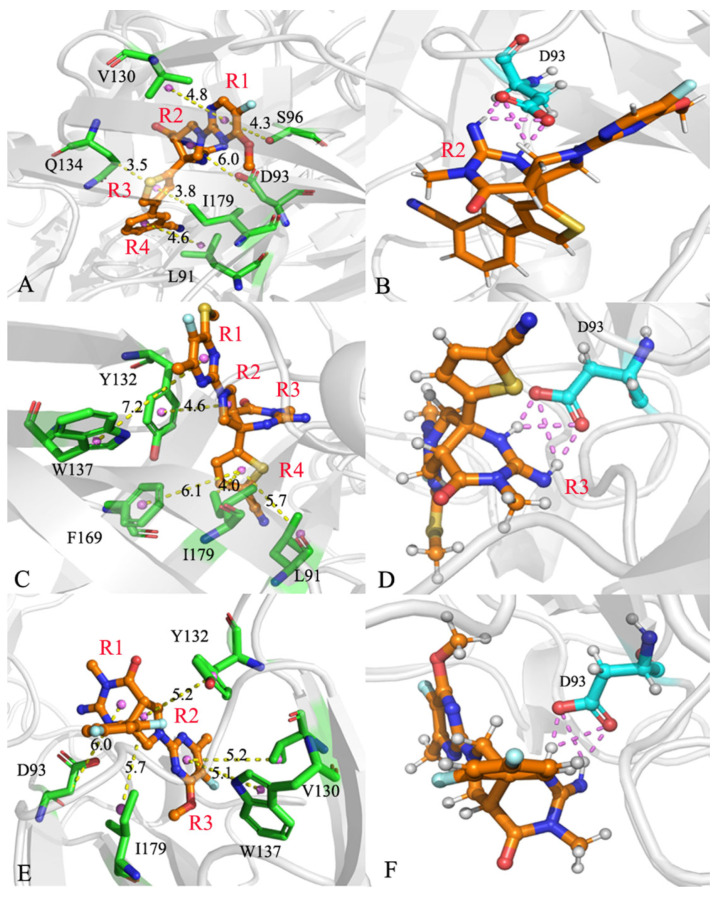
Geometric information of inhibitor–residue interactions; (**A**) the hydrophobic interact ions of 60W with residues, (**B**) the 60W-BACE1 HBIs, (**C**) the hydrophobic interactions between 954 and residues, (**D**) the 954-BACE1 HBIs, (**E**) the hydrophobic interactions of 60X with residues and (**F**) the 60X-BACE1 HBIs.

**Figure 9 molecules-28-04773-f009:**
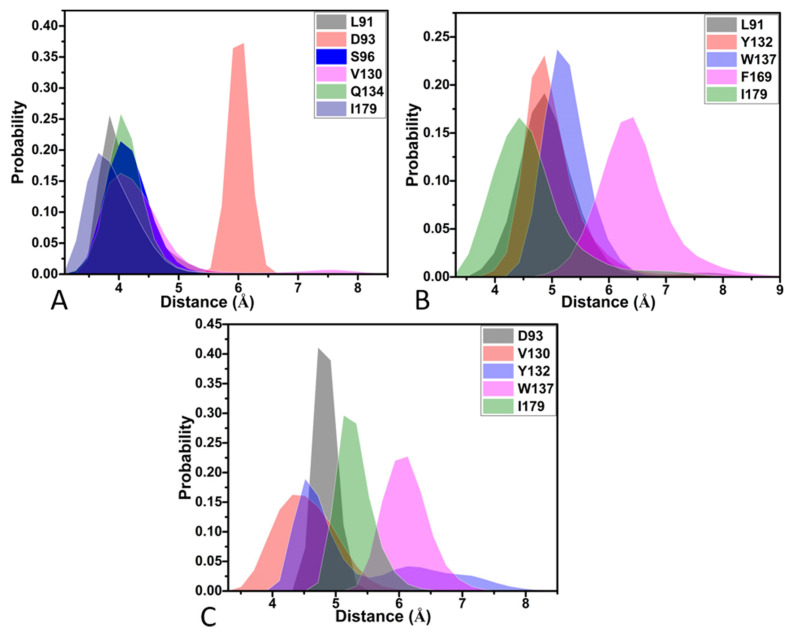
Probability distributions of the distances relating to key inhibitor–residue interactions: (**A**) the 60W-BACE1 complex, (**B**) the 954-BACE1 complex and (**C**) the 60X-BACE1 complex.

**Table 1 molecules-28-04773-t001:** Binding free energies of inhibitors to BACE1 calculated using SIE method ^a^.

Components	60W-BACE1		954-BACE1		60X-BACE1	
	Average	StdErr	Average	StdErr	Average	StdErr
$\Delta E_{vdW}$	−54.53	0.50	−42.68	0.97	−42.50	0.43
$\Delta E_{C}$	−16.18	0.35	−12.55	0.42	−15.22	0.29
$\Delta G^{R}$	24.77	0.37	21.27	0.57	22.80	0.37
γ×ΔMSA	−10.63	0.09	−8.14	0.18	−8.05	0.06
C	−2.89	0.00	−2.89	0.00	−2.89	0.00
^b^ $\Delta G_{bind}$	−8.82	0.07	−7.30	0.11	−7.39	0.05
^c^ $\Delta G_{exp}$	−12.3		−10.0		−11.4	

^a^ All energy components are scaled in kcal·mol^−1^; ^b^
$\Delta G_{bind}=\alpha \times \left[E_{c}+\Delta G^{R}+E_{vdW}+\gamma \times \Delta MSA\right]+C$; ^c^ $\Delta G_{exp}\;$ is obtained using ΔG=−RTlnKi with the experimental value of  Ki [63].

**Table 2 molecules-28-04773-t002:** The parameters used in MM-GBSA calculations with different generalized Born models.

Parameters	IGB = 1	IGB = 2	IGB = 5	IGB = 66
^a^ γ	0.0072	0.005	0.005	0.005
^a^ β	0.00	0.00	0.00	0.00
^b^ radii	mbondi	mbondi2	mbondi2	bondi

^a^ Two empirical parameters used calculations of MM-GBSA. ^b^ Radius type used in selections of GB model, including mbondi, mbondi2 and bondi.

**Table 3 molecules-28-04773-t003:** Binding free energies calculated using MM-GBSA method with different GB models ^a^.

Energy	60W				954				60X			
	IGB = 1	IGB = 2	IGB = 5	IGB = 66	IGB = 1	IGB = 2	IGB = 5	IGB = 66	IGB = 1	IGB = 2	IGB = 5	IGB = 66
$\Delta E_{ele}$	−36.23	−36.23	−36.23	−36.23	−28.25	−28.25	−28.25	−28.25	−33.88	−33.88	−33.88	−33.88
$\Delta E_{vdW}$	−54.58	−54.58	−54.58	−54.58	−42.36	−42.36	−42.36	−42.36	−42.31	−42,031	−42.31	−42.31
$\Delta G_{gb}$	46.47	54.13	11.96	69.01	35.94	42.71	5.57	55.77	40.06	46.70	6.31	56.78
$\Delta G_{surf}$	−7.21	−5.00	−5.00	−5.00	−5.64	−3.92	−3.92	−3.92	−5.71	−3.96	−3.96	−3.96
^b^ $\Delta G_{pol}$	10.24	17.9	−24.27	32.78	7.69	14.46	−22.98	27.52	6.18	12.82	−27.57	22.9
^c^ $\Delta G_{hydro}$	−61.79	−59.58	−59.58	−59.58	−48	−46.28	−46.28	−46.28	−48.02	−46.27	−46.27	−46.27
^d^ ΔH	−51.55	−41.68	−83.85	−26.8	−40.31	−31.82	−69.26	−18.76	−41.84	−33.45	−73.84	−23.37
−TΔS	22.52				18.60				18.94			
$\Delta G_{bind}$	−29.03	−19.16	−61.33	−4.28	−21.71	−13.22	−50.66	−0.16	−22.9	−14.51	−54.9	−4.43
^e^ $\Delta G_{exp}$	−12.3				−10.0				−11.4			

^a^ All free energy components are in scaled in kcal/mol. ^b^
$\Delta G_{pol}=\Delta E_{ele}+\Delta G_{gb}$ which is used to describe polar interactions of inhibitors with BACE1. ^c^
$G_{hydro}=\Delta E_{vdW}+\Delta G_{surf}$ which is utilized to signify hydrophobic interactions of inhibitors with BACE1. ^d^
$\Delta H=\Delta G_{pol}+\Delta G_{hydro}$ which is adopted to indicate the enthalpy effect during bindings of inhibitors to BACE1. ^e^
$\Delta G_{exp}\;$ is obtained via ΔG=−RTlnKi with the experimental value of  Ki [63].

**Table 4 molecules-28-04773-t004:** Contributions of the side chains and backbones to inhibitor–residue interactions ^a^.

Inhibitor	Residue	$S_{vdW}$	$B_{vdW}$	$T_{vdW}$	$S_{ele}$	$B_{ele}$	$T_{ele}$	$S_{gb}$	$B_{gb}$	$T_{gb}$	ΔG
60W	L91	−1.35	−0.07	−1.42	0.06	−0.18	−0.12	−0.05	−0.00	−0.05	−1.72
	D93	−0.59	−0.19	−0.78	−16.62	−0.45	−17.08	15.42	0.51	15.93	−2.04
	S96	−1.69	−0.36	−2.05	1.39	−0.24	1.15	−0.47	0.37	−0.10	−1.13
	V130	−1.43	−0.11	−1.54	−0.03	−0.09	−0.12	0.03	0.07	0.10	−1.75
	Y132	−0.04	−0.07	−0.11	−0.02	0.15	0.13	0.03	−0.02	0.01	0.03
	Q134	−2.96	−0.74	−3.70	−0.26	−0.76	−1.02	0.99	1.52	2.51	−2.58
	W137	−1.34	−0.04	−1.38	−0.06	0.03	−0.03	0.59	−0.03	0.56	−0.93
	F169	−0.69	−0.14	−0.83	0.11	−0.14	−0.03	0.20	0.25	0.45	−0.45
	I179	−1.81	−0.21	−2.02	0.17	−0.05	0.12	−0.11	−0.12	−0.23	−2.26
954	L91	−0.95	−0.05	−1.00	0.07	−0.23	−0.16	−0.04	0.15	0.11	−1.14
	D93	−0.37	−0.10	−0.47	−12.67	−0.32	−12.99	12.51	0.28	12.79	−0.77
	S96	−1.47	−0.34	−1.81	1.09	0.22	1.31	−0.13	0.00	−0.13	−0.78
	V130	−0.72	−0.09	−0.81	−0.02	−0.03	−0.05	0.04	0.13	0.17	−0.77
	Y132	−2.32	−0.11	−2.43	−0.99	0.08	−0.91	1.70	0.01	1.71	−1.88
	Q134	−0.23	−0.06	−0.29	−0.36	−0.04	−0.40	0.39	0.06	0.45	0.27
	W137	−1.66	−0.04	−1.70	−0.38	0.03	−0.35	1.14	−0.02	1.12	−1.04
	F169	−1.44	−0.32	−1.76	−0.46	−0.63	−1.09	0.57	0.96	1.52	−1.44
	I179	−1.64	−0.13	−1.77	0.13	−0.02	0.11	−0.12	−0.04	−0.16	−1.91
60X	L91	−0.75	−0.05	−0.79	0.10	−0.19	−0.09	−0.06	0.11	0.05	−0.90
	D93	−0.30	−0.11	−0.41	−15.06	−0.44	−15.5	14.33	0.44	14.77	−1.23
	S96	−1.71	−0.35	−2.06	1.34	−0.04	1.30	0.07	0.24	0.31	−0.61
	V130	−1.14	−0.11	−1.25	−0.06	0.06	0.00	0.07	0.03	0.10	−1.28
	Y132	−2.52	−0.13	−2.65	−0.61	−0.12	−0.73	1.47	0.18	1.65	−2.00
	Q134	−0.12	−0.05.	−0.17	−0.20	−0.04	−0.24	0.24	0.07	0.31	−0.13
	W137	−1.55	−0.04	−1.59	−0.86	0.06	−0.80	1.25	−0.05	1.20	−1.29
	F169	−1.04	−0.21	−1.25	−0.10	−0.30	−0.40	0.35	0.47	0.82	−0.90
	I179	−1.91	−0.24	−2.15	0.19	−0.11	0.08	−0.11	−0.01	−0.12	−2.31

^a^ All energy components are scaled in kcal/mol. $S_{vdW}$ and $B_{vdW}$ separately indicate contributions of the side chains and backbones to van der Waals interactions ($T_{vdW}$) of inhibitors with residues. $S_{ele}$ and $B_{ele}$ respectively correspond to contributions of the side chains and backbones to electrostatic interactions ($T_{ele}$) of inhibitors with residues. $S_{gb}$ and $B_{gb}$ individually represent contributions of the side chains and backbones to inhibitor–residue polar solvation free energies.

**Table 5 molecules-28-04773-t005:** Hydrogen bonds formed between inhibitors and residues analyzed using the CPPTRAJ.

Compound	^a^ Hydrogen bonds	Distance (Å)	Angle (°)	^b^ Occupancy (%)
60W-BACE1	A93–OD1…60W-H3–N1	3.0	157.1	91.1
	A93–OD2…60W-H1–N	2.8	163.3	82.4
	A93–OD2…60W-H3–N1	3.2	147.8	54.1
	A93–OD1…60W-H1–N	3.1	143.8	46.7
954-BACE1	A93–OD2…954-H4–N2	3.1	150.1	76.7
	A93–OD1…954-H4–N2	3.1	149.5	75.6
	A93–OD1…954-H2–N1	2.8	159.6	49.2
	A93–OD2…954-H2–N1	2.9	158.5	46.9
60X-BACE1	A93–OD2…60X-H5–N4	3.1	150.4	89.1
	A93–OD1…60X-H5–N4	3.1	149.3	84.4
	A93–OD1…60X-H1–N3	2.8	161.6	62.7
	A93–OD2…60X-H1–N3	2.9	157.7	48.7

^a^ Hydrogen bonding interactions are recognized by an acceptor···donor distance of <3.5 Å and acceptor···H donor angle of >120°. ^b^ Occupancy (%) is defined as the percentage of the simulation time that a specific hydrogen bond exists.

## Data Availability

Not applicable.

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
