# Peer review of "Identification Mechanism of BACE1 on Inhibitors Probed by Using Multiple Separate Molecular Dynamics Simulations and Comparative Calculations of Binding Free Energies"

_molecules, 2023, doi:10.3390/molecules28124773_

Round 1

Reviewer 1 Report

The authors present a comprehensive account of different order parameters computed from multiple MD simulations of small molecule ligands bound to BACE1 to elucidate their role in the inhibition of BACE1’s downstream signaling pathway that has implications for Alzheimer’s. The study is extremely prolific and the conclusions are solid based on the data presented, and especially based on the well-rounded analyses from different perspectives such as structural order parameters, residue fluctuations and thermodynamic free energy surfaces. I only have minor comments here:

  1. In line 92: “Binding ability of 60W, 954 and 60X to BACE1 is scaled by the Ki values of 1, 45 and 48 nM” → What is the rationale for this scaling and where do the authors obtain these normalizing values from?

  1. In Fig 2A, perhaps just use the word distribution or probability instead of “frequency”. Also, would it make more sense to measure the RMSD of residues around the binding pocket, instead of the whole BACE1 structure, since the RMSD changes overall are extremely small (~ 2 A)? Please also make the colors consistent between Figs 2A and 2B, otherwise it creates a lot of confusion for the reader.

Is it possible to report which conformations of 60W-BACE1 complex correspond to which conformations of free 60W between Figs 2A and B, since both these distributions have two distinct peaks?

In Fig 2C, the differences are impossible to decipher. Please enhance the clarity here.

  1. I suggest the authors discuss somewhere what a better objective for drug design is in this case: more stable small molecule binders (stability as discussed through binder RMSD as in Fig. 2), or more stable structure of the receptor pocket, and correlate it with what they find in this study. 

  1. I have some confusion between the conclusions of Fig. 3C and Fig 4. In Fig 3C, the radius of gyration (Rg) of inhibitor bound BACE1 collapses to a single peak than the apo BACE1, which the authors claim as demonstrating an increase in compactness on ligand binding. However, the difference in the peak value of the Rgs are 1-2 A which is significantly smaller than the overall size of BACE1. Do the authors still think that their simulations really show a change in “compactness”? If anything, the Rgs increase from apo to bound BACE1. 

On the other hand, Fig 4 (B)-(D) reflects a decay of dynamic correlations in ligand bound BACE1. So it looks like ligand binding makes BACE1 more “rigid” (and perhaps more thermostable) than “compact”. While this looks like just a change of words, the intended scientific message also completely changes.

  1. On a related note, in Eq. (4), the correlation coefficient is calculated from CA displacements which are un-normalized. How does the final value of the coefficient then automatically become normalized between -1 and 1. Please provide a correction or a clarification.

Author Response

Dear Editor,

First of all, we would like to thank you for giving us the opportunity to revise our manuscript.

According to valuable suggestions from reviewers, we have revised our manuscript. Our point-by-point responses to the comments of the reviewers and editors were attached below.

Thanks a lot for your kind help and suggestions.

Yours Sincerely

Weikai He

Response to reviewer 1:

Thank you very much for your valuable suggestions and comments in advance.

  1. The authors present a comprehensive account of different order parameters computed from multiple MD simulations of small molecule ligands bound to BACE1 to elucidate their role in the inhibition of BACE1’s downstream signaling pathway that has implications for Alzheimer’s. The study is extremely prolific and the conclusions are solid based on the data presented, and especially based on the well-rounded analyses from different perspectives such as structural order parameters, residue fluctuations and thermodynamic free energy surfaces. I only have minor comments here.

Author reply:

Thank you very much for your positive comments.

  1. In line 92: “Binding ability of 60W, 954 and 60X to BACE1 is scaled by the Ki values of 1, 45 and 48 nM” → What is the rationale for this scaling and where do the authors obtain these normalizing values from?

Author reply:

       Thank you very much for this valuable comment.

       According to avoid mistake, we have revised this description and the changes are marked by using the “Track Changes” function in MS Word.

       Thank you again for your valuable comment.

  1. In Fig 2A, perhaps just use the word distribution or probability instead of “frequency”. Also, would it make more sense to measure the RMSD of residues around the binding pocket, instead of the whole BACE1 structure, since the RMSD changes overall are extremely small (~ 2 A)? Please also make the colors consistent between Figs 2A and 2B, otherwise it creates a lot of confusion for the reader.

Author reply:

       Thank you very much for this valuable suggestion

       By following your valuable suggestion, Figure 2A and 2B have been revised and the corresponding contents have been also revised. The RMSDs of the binding pocket, namely residues near the 7 Å away from the mass center of inhibitors, are have been calculated, which is used to replace the previous Figure 2A.

       Thank you again for your valuable suggestion.

  1. Is it possible to report which conformations of 60W-BACE1 complex correspond to which conformations of free 60W between Figs 2A and B, since both these distributions have two distinct peaks?.

Author reply:

       Thank you very much for this valuable suggestion.

       The whole RMSDs have been revised as that of the binding pocket. The new RMSDs show the single distribution. The results better imply the effect of inhibitor binding on the binding pocket of BACE1. Based on your valuable suggestion, we have revised our manuscript and the revised contents are marked through the “Track Changes” function in MS Word.

       Thank you again for this valuable suggestion.

  1. In Fig 2C, the differences are impossible to decipher. Please enhance the clarity here.

Author reply:

       Thank you very much for this valuable suggestion.

       Based on your valuable suggestion, we have revised Figure 2C and the corresponding description. The revised parts are highlighted through the “Track Changes” function in MS Word.

       Thank you again for your valuable suggestion.

  1. I suggest the authors discuss somewhere what a better objective for drug design is in this case: more stable small molecule binders (stability as discussed through binder RMSD as in Fig. 2), or more stable structure of the receptor pocket, and correlate it with what they find in this study.

Author reply:

       Thank you very much for your valuable suggestion.

       By following your valuable suggestion, we have revised our manuscript and the revised parts are marked with the “Track Changes” function in MS Word.

       Thank you again for this valuable suggestion.

  1. I have some confusion between the conclusions of Fig. 3C and Fig 4. In Fig 3C, the radius of gyration (Rg) of inhibitor bound BACE1 collapses to a single peak than the apo BACE1, which the authors claim as demonstrating an increase in compactness on ligand binding. However, the difference in the peak value of the Rgs are 1-2 A which is significantly smaller than the overall size of BACE1. Do the authors still think that their simulations really show a change in “compactness”? If anything, the Rgs increase from apo to bound BACE1.

Author reply:

       Thank you very much for your valuable comment.

       We also agree with you, binding of inhibitors only brings slight effect on compact extents of BACE1. Hence, according to your valuable suggestion, we have revised our description on gyrations. The revised parts are marked though the “Track Changes” function in MS Word.

  1. On the other hand, Fig 4 (B)-(D) reflects a decay of dynamic correlations in ligand bound BACE1. So it looks like ligand binding makes BACE1 more “rigid” (and perhaps more thermostable) than “compact”. While this looks like just a change of words, the intended scientific message also completely changes.

Author reply:

       Thank you very much for this valuable comment.

       DCCMs only reflect the changes in correlated motions between the structural regions. There is not direct relationship between the changes in correlated motions and the structural compact extents. Based on your valuable comment, we have revised our manuscript and the revised parts are highlighted through the “Track Changes” function in MS World.

  1. On a related note, in Eq. (4), the correlation coefficient is calculated from CA displacements which are un-normalized. How does the final value of the coefficient then automatically become normalized between -1 and 1. Please provide a correction or a clarification.

Author reply:

       Thank you very much for this valuable suggestion.

       The Eq. (4) stems from the work of Ichiye et al. and the corresponding references have been provided. The values -1 and 1 can be obtained when i=j. When i is different from j, the absolute values of the correlation coefficient is smaller than the absolute value of -1 and 1.

       Finally, thank you again for your valuable suggestions and efforts paid by you in reviewing our manuscript.

Reviewer 2 Report

Authors studied the topic "Identification mechanism of BACE1 on inhibitors probed by using multiple separate molecular dynamics simulations and comparative calculations of binding free energies" through computational method.

1.In Fig 6. of free energy landscape authors recheck the representation of Fig 6 C, whether it is EB1/EB2 or EV1/EV2.

2. In Table 1. the unit of binding free energies of inhibitors is in proper super script. (kcal. mol-1) .

3. Besides residue based energy the authors expand the conclusion section more.

Good

Author Response

Dear Editor,

First of all, we would like to thank you for giving us the opportunity to revise our manuscript.

According to valuable suggestions from reviewers, we have revised our manuscript. Our point-by-point responses to the comments of the reviewers and editors were attached below.

Thanks a lot for your kind help and suggestions.

Yours Sincerely

Weikai He

Response to the Reviewer2

Thank you very much for your valuable suggestions and comments in advance

  1. In Fig 6. of free energy landscape authors recheck the representation of Fig 6 C, whether it is EB1/EB2 or EV1/EV2.

Author reply:

Thank you very much for your valuable suggestion.

We have corrected this mistake and are sorry for this issue.

Thank you again for this reminding.

  1. In Table 1. the unit of binding free energies of inhibitors is in proper super script. (kcal. mol-1) .

Author reply:

       Thank you very much for this kindly reminding

       The unit has been revised.

  1. Besides residue based energy the authors expand the conclusion section more.

Author reply:

       Thank you very much for this valuable suggestion.

       The contents of the conclusion section has been expanded.

       Finally, thank you very much for your valuable suggestion and efforts paid by you in reviewing our manuscript.

Reviewer 3 Report

The authors present a detailed computational characterization of the interaction between BACE1 protein and three inhibitors. In general, the work will be of interest to those working in computer aided drug design and in Alzheimer's disease drug discovery. The type of analysis performed gives information at atomic level about the hot spots to design new inhibitors against BACE1 protein. Just some observations.

1.- Why do not to include inhibitors with different structure? please discuss this in the text. 

2.- In the same context, what about the interactions described for other inhibitors?

3.- A deeper discussion related with the data in literature is needed.

Author Response

Dear Editor,

First of all, we would like to thank you for giving us the opportunity to revise our manuscript.

According to valuable suggestions from reviewers, we have revised our manuscript. Our point-by-point responses to the comments of the reviewers and editors were attached below.

Thanks a lot for your kind help and suggestions.

Yours Sincerely

Weihai He

Response to reviewer 3:

Thank you very much for your valuable suggestions and comments in advance.

  1. Why do not to include inhibitors with different structure? please discuss this in the text.

Author reply:

Thank you very much for this valuable suggestion.

       According to your valuable suggestion, we have revised our manuscript in introduction and the revised contents are marked with the “Track Changes” function in MS World.

       Thank you again for this valuable suggestion.

  1. In the same context, what about the interactions described for other inhibitors?

Author reply:

       Thank you very much for this valuable suggestion.

       According to your valuable suggestion, we have revised our manuscript and the revised contents are highlighted through the “Track Changes” function in MS Word.

  1. A deeper discussion related with the data in literature is needed.

Author reply:

       Thank you very much for this valuable suggestion

       By following your valuable suggestion, the related discussion has been added. The revised parts are indicated through the “Track Changes” function in MS Word.

       Finally, thank you again for your valuable suggestions and efforts paid by you in reviewing our manuscript.

Reviewer 4 Report

The authors have done a detailed study on molecular dynamics of inhibitors of BACE1, which will help in designing a suitable drug for AD.

The article is well presented, I have a few minor comments for improvement

1. Please improve the English language throughout the paper.

2. The title may be changed to-‘Molecular dynamic simulations provide insights into the binding mechanism of inhibitors to BACE1’.

3. Conclusion should also mention which inhibitor was better.

Some corrections are marked in the pdf.

Please improve the English language throughout the paper.

Some corrections are marked in the pdf.

Author Response

Dear Editor,

First of all, we would like to thank you for giving us the opportunity to revise our manuscript.

According to valuable suggestions from reviewers, we have revised our manuscript. Our point-by-point responses to the comments of the reviewers and editors were attached below.

Thanks a lot for your kind help and suggestions.

Yours Sincerely

Weikai HE

Response to reviewer 4:

Thank you very much for your valuable suggestions and comments in advance.

  1. Please improve the English language throughout the paper.

Author reply:

Thank you very much for this valuable suggestion.

       We have revised our English description through our manuscript with the help of an expert whose mother language is English. The changes have been highlighted through the “Track Changes” function in MS word.

  1. The title may be changed to-‘Molecular dynamic simulations provide insights into the binding mechanism of inhibitors to BACE1’.

Author reply:

       Thank you very much for this valuable suggestion.

       The title has been revised as “Molecular dynamic simulations provide insights into the binding mechanism of inhibitors to BACE1”. The changes have been highlighted through the “Track Changes” function in MS word.

  1. Conclusion should also mention which inhibitor was better.

Author reply:

       Thank you very much for this valuable suggestion

       According to your valuable suggestion, the performance of inhibitors has been mentioned in conclusion. The changes have been highlighted through the “Track Changes” function in MS word.

  1. Some corrections are marked in the pdf.

Author reply:

       Thank you very much for these valuable suggestions.

       Some corrections marked in your pdf have been performed. The changes have been highlighted through the “Track Changes” function in MS word.

Finally, thank you very much for your valuable suggestion and efforts paid by you in reviewing our manuscript.

Reviewer 5 Report

The authors should address the following comments:

1-in Page 4, lines 147-149: why did the authors use the "stability of secondary structure"? it is recommended to use "alteration in the secondary structure" because in the field of protein stability, the alterations in the tertiary structure and the equilibrium between the fold and unfold states of the proteins are considered rather than secondary structure.

2-In Page 3, line93: The dimension of the binding constant is nM!! is that related to the dissociative binding constant?? It seems that it should be associative binding constant as shown in the equations in the legends of the Tables of the binding energies.

3- inPage 3, lines 86-88: It is recommended to rewrite the sentences.  

Author Response

Dear Editor,

First of all, we would like to thank you for giving us the opportunity to revise our manuscript.

According to valuable suggestions from reviewers, we have revised our manuscript. Our point-by-point responses to the comments of the reviewers and editors were attached below.

Thanks a lot for your kind help and suggestions.

Yours Sincerely

Jianzhong Chen

Response to reviewer 5:

Thank you very much for your valuable suggestions and comments in advance.

  1. 1-in Page 4, lines 147-149: why did the authors use the "stability of secondary structure"? it is recommended to use "alteration in the secondary structure" because in the field of protein stability, the alterations in the tertiary structure and the equilibrium between the fold and unfold states of the proteins are considered rather than secondary structure.

Author reply:

Thank you very much for this valuable suggestion.

       According to your valuable suggestion, we have revised our manuscript and the revised parts are marked through the “Track Changes” function in MS Word.

  1. In Page 3, line93: The dimension of the binding constant is nM!! is that related to the dissociative binding constant?? It seems that it should be associative binding constant as shown in the equations in the legends of the Tables of the binding energies.

Author reply:

       Thank you very much for this valuable suggestion.

       According to your valuable suggestion, we have revised our manuscript and the changed contents are marked through the “Track Changes” function in MS Word.

       Thank you again for your valuable suggestion.

  1. inPage 3, lines 86-88: It is recommended to rewrite the sentences.

Author reply:

       Thank you very much for this valuable suggestion

       By following your valuable suggestion, the corresponding contents have been revised and the evised parts have been highlighted with the red.

       Thank you again for your kindly reminding.

       Finally, thank you very much for your valuable suggestion and efforts paid by you in reviewing our manuscript.
